# Dual-comb thin-disk oscillator

Kilian Fritsch [1✉], Tobias Hofer[1], Jonathan Brons[2,3], Maksim Iandulskii[2], Ka Fai Mak [4], Zaijun Chen[4], Nathalie Picqué [4] & Oleg Pronin[1]

Dual-comb spectroscopy (DCS) normally operates with two independent, relatively low power and actively synchronized laser sources. This hinders the wide adoption for practical implementations and frequency conversion into deep UV and VUV spectral ranges. Here, we report a fully passive, high power dual-comb laser based on thin-disk technology and its application to direct frequency comb spectroscopy. The peak power (1.2 MW) and the average power (15 W) of our Yb:YAG thin-disk dual-comb system are more than one-order-of-magnitude higher than in any previous systems. The scheme allows easy adjustment of the repetition frequency difference during operation. Both combs share all cavity components which leads to an excellent mutual stability. A time-domain signal recorded over 10 ms without any active stabilization was sufficient to resolve individual comb lines after Fourier transformation.

[1] Helmut-Schmidt-Universität/Universität der Bundeswehr Hamburg, Holstenhofweg 85, D-22043 Hamburg, Germany. [2] Ludwig-Maximilians-Universität München, Am Coulombwall 1, D-85748 Garching, Germany. [3] TRUMPF Laser GmbH, Aichhalder Str. 39, D-78713 Schramberg, Germany. [4] Max-Planck-Institut für Quantenoptik, Hans-Kopfermann-Str. 1, D-85748 Garching, Germany. ✉email: kilian.fritsch@hsu-hh.de

Fourier-transform spectrometer[1] measures the time-domain interference between two waves while their relative phase-shift is varied. The Fourier transform of the interference pattern provides the spectrum. Advanced implementations use lasers as light sources. Their increased brightness compared to incoherent sources leads to a shorter measurement time or an increased signal-to-noise ratio (SNR)[2]. Even further refined systems incorporate optical frequency combs (OFC) as light sources. This enhances the spectral resolution, which is ultimately limited by the OFCs individual comb linewidth; improves sensitivity[3]; and reduces the effect of instrument line shape on the measurement[2]. Experiments comprising two OFCs, known as dual-comb spectroscopy (DCS), emulate the mechanically scanned delay in Fourier-transform spectroscopy by two separate pulse trains with slightly different repetition frequencies (asynchronous pulse trains). DCS is rapidly advancing since it was first proposed[4] and demonstrated in a Ti:Sapphire-driven mid-infrared system[5]. The key reason is the simple and fast electronic detection of the interferogram generated by two heterodyned asynchronous pulse trains on a photodetector or by means of electro-optic sampling[6]. Advantages arise in terms of acquisition speed, spectral resolution, accuracy, robustness, and SNR over traditional scanning Fourier-transform spectroscopy. The principle is extendable to nearly all related spectroscopy methods, and several fundamental experimental demonstrations have been published. They included Doppler-free spectroscopy[7], anti-Stokes Raman-spectroscopy[8], and two-photon spectroscopy[9], as well as scattering-type scanning near-field optical microscopy[10,11]. DCS applications include trace gas detection, lidars, and kinetics of chemical reactions, as reviewed in [2] and [12].

One major challenge with DCS is implementing an interferometer with two combs. They must be mutually coherent and operate at different repetition rates[12], whilst spanning the necessary bandwidth in the examined spectral region. One promising way of realizing such a light source is by electronically modulating a continuous wave laser with two intensity modulators at different frequencies[13]. More commonly though, the two OFCs are generated by separate mode-locked femtosecond oscillators running at slightly offset repetition rates. This involves the complex active stabilization of repetition rates and carrier-envelope offset frequencies or their active tracking and a posteriori correction. To date, erbium fiber lasers[14] show the best performance, based on active stabilization or a posteriori correction schemes.

An emerging way is to generate a dual-comb output from a single laser cavity. This could simplify the technique, as many noise sources are expected to be shared by both pulse trains. This alleviates the need for active stabilization over short time periods. So far, there have been only a few demonstrations of such a scheme. They rely on different effects to generate a repetition frequency offset between the two outputs. For example, polarization multiplexing in MIXSELs[15], in bulk oscillators[16] and chromatic dispersion in a dual-wavelength fiber laser system[17] have been demonstrated. In bulk solid-state lasers, the directional dependence of the Kerr nonlinearity[18,19] was exploited. A selection of passive dual-comb lasers is given in Supplementary Table 1. It compares average and peak powers, repetition rates, differences in repetition rates, pulse durations, spectral spans, central wavelengths and types of laser technology. Average and peak powers are particularly important for spectral broadening, pulse compression and spectral extension into the THz, infrared (IR), ultraviolet (UV) and vacuum ultraviolet (VUV) spectral regions where most of the strong and spectroscopically relevant transitions are situated. The conversion efficiency, which is intrinsically low for these nonlinear processes, must be compensated by a high photon flux driving laser.

In this work, we present a thin-disk based dual-comb femtosecond oscillator. Thanks to the thin-disk platform[20], the presented system shows more than one order of magnitude higher output average and peak powers compared to previous dual-comb demonstrations. No noise is added by external optically active media, since the thin-disk laser provides enough power without subsequent amplification. The scheme effectively rejects common noise without active stabilization. We report a broad tuning range of the difference of the repetition frequencies $\Delta f_{\rm rep}$ during mode-locked laser operation by simply adjusting the relative longitudinal offset of the end mirrors. In contrast to optical systems based on a single cavity in Supplementary Table 1, this allows optimal and flexible choice of the important parameter $\Delta f_{\rm rep}$.

Given its output parameters, it represents a relatively simple and compact table-top source for DCS, particularly promising for further nonlinear conversion into the IR, UV and VUV spectral regions and realization of different nonlinear spectroscopy schemes.

## Results

The experimental setup (Fig. 1) is a high-power Kerr-lens mode-locked (KLM) Yb:YAG thin-disk oscillator pumped by a 500 W-class fiber-coupled diode laser. It features a thin-disk module consisting of the disk and the pump cavity made by TRUMPF and high-power optimized dispersive mirrors designed and manufactured by Dr V. Pervak and his team. The dual layout of this thin-disk laser contains two nearly identical laser resonators, which share all cavity mirrors, including the active area on the thin-disk (Fig. 1c, pump spot). An exception to this principle constitutes the end mirrors and output couplers. They are split horizontally to allow the alignment of the two cavities independently[21]. The spatial separation is roughly 5 mm in the vertical (sagittal) plane (Fig. 1a, separation drawn horizontally (tangentially) for simplified visualization). The spatial separation allows individual fine tuning of the repetition frequencies. Their difference $\Delta f_{\rm rep}$ is a key parameter in any DCS systems. It

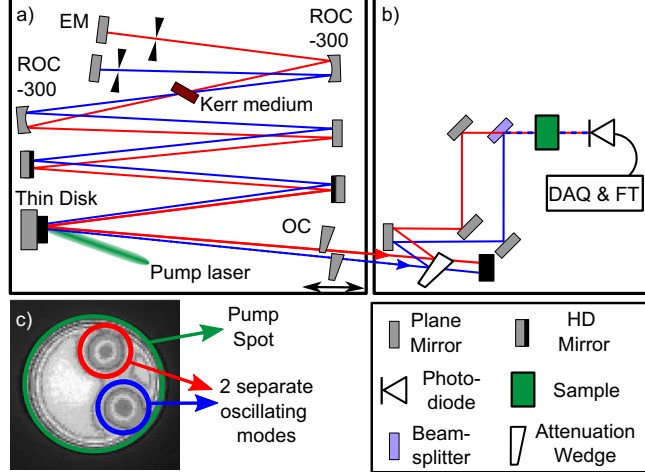

**Fig. 1 Schematic drawing of the dual-comb thin-disk laser system. a** The laser is based on the Yb:YAG thin-disk resonator. The beam-paths corresponding to the "red" and "blue" beams are separated in tangential plane for illustration only. EM end mirror, ROC radius of curvature, OC output coupler (10 % transmission). **b** The heterodyne detection setup. DAQ data acquisition board, FT Fourier transform. **c** Image of the pump spot (laser active area) on the thin disk. Both laser modes share a single pump spot and therefore a single pump laser. They show a slight horizontal displacement that is a result of the optimized alignment.

determines the dead-time between consecutive interferograms and the line-spacing of the down-sampled RF comb[22]. In the presented arrangement, $\Delta f_{\text{rep}}$ can be easily tuned, up to the kilohertz range, by longitudinally translating one of the split output couplers. The tuning of $\Delta f_{\text{rep}}$ up to 1 MHz was experimentally verified. This simple tunability of $\Delta f_{\text{rep}}$ can allow the implementation of fast scanning methods, such as those that rely on the inversion of the sign of $\Delta f_{\text{rep}}$, demonstrated in[23]. This is a major advantage over DCS systems that rely on different methods to introduce $\Delta f_{\text{rep}}$, i.e., via birefringence[24] or chromatic dispersion[17]. Except for[25,26], those systems are quite limited in tunability and their tunability range (see Supplementary Table 1 for more information), because changes involve rearranging or realigning the cavities as opposed to simply translating a mirror.

The configuration of an individual cavity, without the stacked dual-cavity feature, was inspired by Brons et al.[27]. A telescope comprising two concave mirrors with a radius of curvature of 300 mm focuses both cavity modes into two distinct foci within a Kerr medium composed of a 4 mm sapphire plate placed under Brewster's angle. Radial restriction of the modes by hard apertures and soft aperture effects in the gain medium combined with the self-focusing in the sapphire plate enables KLM[28]. Despite the vertical displacement of the laser beams, it is possible to use a single sapphire plate to mode-lock both cavities. Similar to conventional KLM thin-disk oscillators, both cavities can be mode-locked at the same time by manually perturbing one of the telescope mirrors. To achieve soliton mode-locking in the net-negative dispersion regime, two highly dispersive mirrors, compensating a Group Delay Dispersion of approximately $-3000\ \text{fs}^2$ per reflection, resulting in $-12,000\ \text{fs}^2$ total intra-cavity dispersion per round trip, are placed inside the resonators.

The common-path cavity design ensures temporal stability of the dual-output system as well as a reasonable mutual coherence time between the two frequency combs in the absence of any active stabilization mechanisms. The two cavities share nearly the same intra-cavity optical components and the same pump source. Thus, they experience nearly equal perturbations originating from air flows, mechanical vibrations, and the intensity noise of the pump source. However, the system can be somewhat more sensitive to the drifts and vibrations associated with the displacement of the beams in a plane parallel to the optical table.

**Output characteristics of the dual-comb system.** The spectra of both pulse trains, measured with an optical spectrum analyzer (OSA, Yokogawa), are compared in Fig. 2. They show almost identical spectral shapes and widths, indicating almost identical output pulses. Both spectra are centered at 1030 nm, and they have slightly different average output powers of 16 and 15 W, which corresponds to 265 and 250 nJ per pulse, respectively. This yields a combined optical-to-optical efficiency of 12% at the 270 W pump power. The repetition frequency $f_{\text{rep}}$ was set to 60.4 MHz. The $\text{sech}^2$-shaped output pulses have 176 and 186 fs FWHM pulse durations, measured with a commercial autocorrelator (APE). The peak powers are calculated to be around 1.32 and 1.2 MW, respectively. To the best of our knowledge, the achieved peak power exceeds the currently reported highest peak power from a single laser dual-comb system of 0.03 MW[18] by approximately 40 times. It is also more than 18 times larger than the currently reported highest peak power of 0.07 MW from a dual-comb CPA system[29]. The small differences between the two outputs can arise from the fact that the optical axes are not identical resulting in different angles of incidence on the telescope mirrors and Kerr medium. This can be mitigated easily by attenuating the more powerful beam to match the lower power one. The different beam positions on the thin-disk can also result

in a slight gain difference between the two lasing modes. Unequal amounts of defects on the mirrors, as well as a mismatch in the size of the hard apertures, also can create differences in linear losses.

The time evolution of the two repetition frequencies and their difference is shown in Fig. 3. Although both repetition frequencies decrease monotonically with time, their difference $\Delta f_{\text{rep}}$ fluctuates only weakly with a standard deviation of 0.1 Hz on a minute time-scale. This may suggest that $\Delta f_{\text{rep}}$ is also fluctuating with a similar standard deviation on a millisecond time scale, the typical acquisition times for fast DCS experiments. Careful analysis of the DCS time trace (see Fig. 4a) shows that $\Delta f_{\text{rep}}$ fluctuations (see Supplementary information for details) are non-negligible, and it confirms the standard deviation of approximately 0.1 Hz. The $\Delta f_{\text{rep}}$ fluctuations are happening on the time scale 10–100 ms (see Supplementary Fig. 1). The presented stability level (in Fig. 3) is comparable to previous publications[17,18,30,31], which is remarkable considering the at least one order of magnitude higher output average powers and intra-cavity power of approximately 150 W. Further active stabilization of the difference in repetition frequencies should not be required, but it could be implemented for more challenging experiments. A posteriori numerical correction of $\Delta f_{\text{rep}}$ fluctuations is desirable, and it significantly improves the results as shown further in text. Furthermore, the pulse energy is over one order of magnitude higher than the 7.7 nJ used in the CPA system in[29] and it far exceeds the energies delivered by state-of-the-art single laser systems. In future versions of the presented

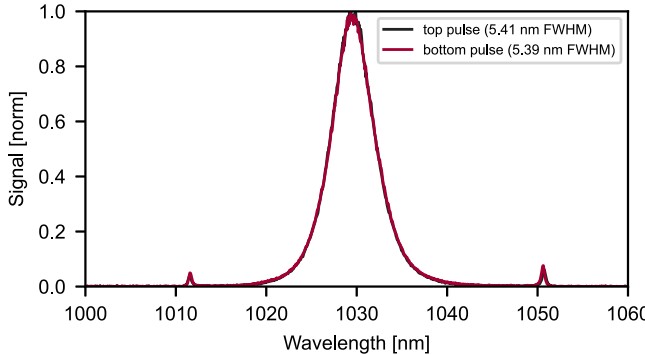

**Fig. 2 Spectral characteristics of the emitted pulses.** In black, the spectrum of "top pulses", marked with red circle in Fig. 1c, is displayed. In red, the spectrum of "bottom pulses", marked in Fig. 1c with a blue circle, is shown.

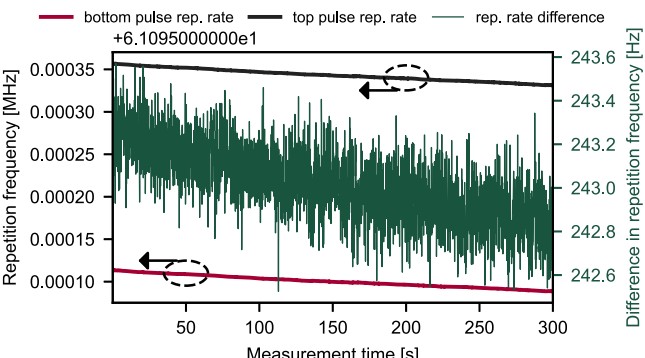

**Fig. 3 Temporal stability of the repetition rates and the difference in the repetition rates.** Both repetition frequencies monotonically decrease with time (black and red curves).

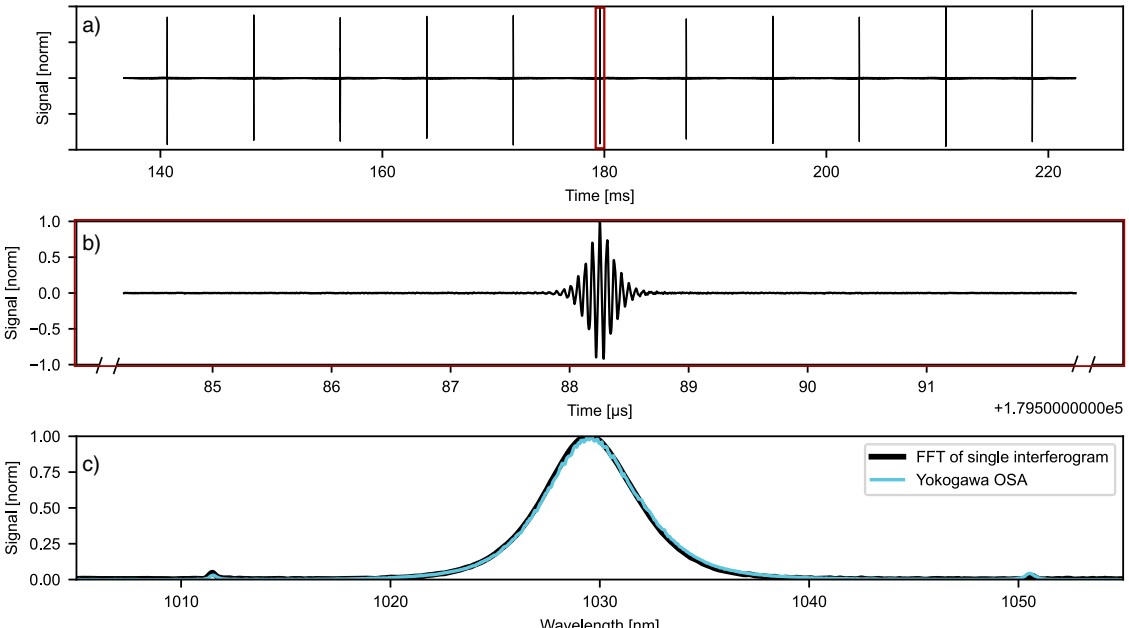

**Fig. 4 Measurements with the dual-comb system without any sample. a** Time-trace with 11 consecutive interferograms. Time separation between neighboring interferograms is 7.8 ms. **b** Single interferogram from center in (**a**). **c** FFT spectrum retrieved from a single 20 μs time-trace shown in (**b**) and OSA reference spectrum. The spectral features (Kelly sidebands) around 1012 and 1052 nm are reproduced very well that confirms accuracy for a spectral span of $\Delta\lambda = 40$nm.

system, the pulse energy could be scaled up easily as described by Brons et al.[20] by increasing the focal length of the cavities curved mirrors. For example, a set of −600 mm mirrors instead of −300 mm mirrors should yield a twofold increase.

**Dual-comb measurements.** By combining two laser beams on a photodiode (see setup in Fig. 1b), interferograms are recorded in the time domain. Fig. 4a shows a 100 ms trace with 11 consecutive interferograms. $\Delta f_{rep}$ was adjusted to 128 Hz, which corresponds to 7.8 ms temporal separation between consecutive interferograms and a down-conversion factor of $a = \frac{\Delta f_{rep}}{f_{rep}} = 2.12 * 10^{-6}$[22]. The optical spectrum (Fig. 4c) can be obtained by taking a Fourier transform of an individual interferogram centered in a 20-μs time-window (Fig. 4b) and scaling it appropriately by the down-conversion factor. No sample was used in this measurement. The spectrum agrees well with a reference spectrum recorded by an OSA (Yokogawa). Characteristic Kelly sidebands of our oscillators at 1012 and 1052 nm are clearly visible in Fig. 4c, which confirms spectral accuracy of our DCS laser source for spectral spans as large as 40 nm.

To investigate fluctuations of $\Delta f_{rep}$, the arrival time of the interferograms and their jitter are evaluated in time domain (see Supplementary information). This analysis showed that the correcting the jitter of the repetition rate difference (or burst arrival time) is crucial. However, we still assume that $\Delta f_{rep}$ remains constant on the microsecond timescale and therefore cannot affect the width of the spectra during measurement. This may indicate that the dominating instabilities are due to fluctuations in the repetition rate difference ($\Delta f_{rep}$). They can be compensated numerically using a simple post-processing technique (see Supplementary information for details).

To resolve the individual comb lines, the measurement time must be increased to yield enough data to enhance frequency resolution. Furthermore, the increase in the acquisition time will improve the SNR if the two frequency combs stay coherent. After correcting the interferograms, which only involves corrections of the difference in repetition frequencies, the SNR scales as $\sim \sqrt{T}$ where $T$ is the measurement time (see Supplementary information).

Figure 5a depicts optical spectra calculated from a trace containing 40 interferograms. A comparison between the spectra with apodization and the spectra without any correction shows that the $\text{SNR}_{apod.} = 1417$ for the apodized case is significantly improved compared to the spectra without any correction with $\text{SNR} = 25$. The huge increase in SNR and cleaning up of the spectrum is due to the relatively narrow window (2 μs) applied during apodization.

Figure 5b depicts optical spectra with the $\Delta f_{rep}$ correction technique noted above applied, including apodization and the spectra without any correction. This case shows $\text{SNR}_{time\ corr.-apod.} = 3112$ compared to $\text{SNR} = 25$ of the non-corrected case while the good agreement with the reference measurement remains unchanged. The inset of Fig. 5b shows that the spectral resolution is already sufficiently high to resolve the comb structure of the optical spectrum.

The 312 ms time trace with 40 consecutive interferograms was analyzed after applying the $\Delta f_{rep}$ correction. The analysis showed that the FWHM of a single comb line (Fig. 6) of the spectrum is 3.1 Hz in the RF domain. Interleaving multiple measurements can bring the spectral resolution close to the width of the comb line in the optical domain[32]. Moreover, using the aforementioned $\Delta f_{rep}$ correction technique on arbitrarily long measurements containing a vast number of interferograms can improve the SNR even further. With minor corrections to the $\Delta f_{rep}$ only, a time-domain trace of 312 ms can be transformed without artifacts, increasing a posteriori the effective coherence time by simple numerical means[33].

In addition, we performed a spectroscopic measurement of the transmission of a home-built Fabry–Pérot etalon with a free spectral range of 140 GHz and an FWHM-linewidth of 24 GHz. The reference spectrum (OSA) and the measured (retrieved)

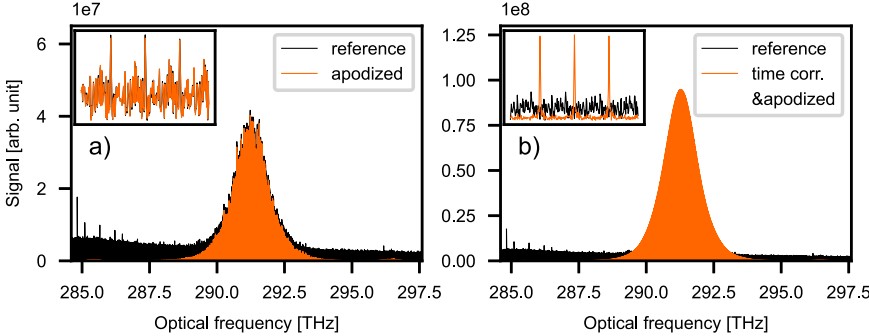

**Fig. 5 Dual-comb spectra before and after applying the correction. a** Apodized (orange) and not corrected (black) spectra transformed from a 312 ms time trace. Forty interferograms contributed in both cases. It is clearly seen that $\Delta f_{rep}$ jitter-corrected spectrum experiences a significant decrease in noise. The inset shows the frequency comb lines of both spectra. **b** $\Delta f_{rep}$ jitter-corrected and apodized spectrum and uncorrected reference.

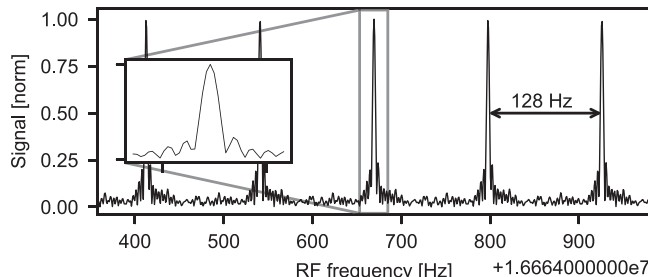

**Fig. 6 Comb line structure of the optical spectrum.** The inset shows a single comb line with FWHM of 3.1 Hz in radio frequency domain.

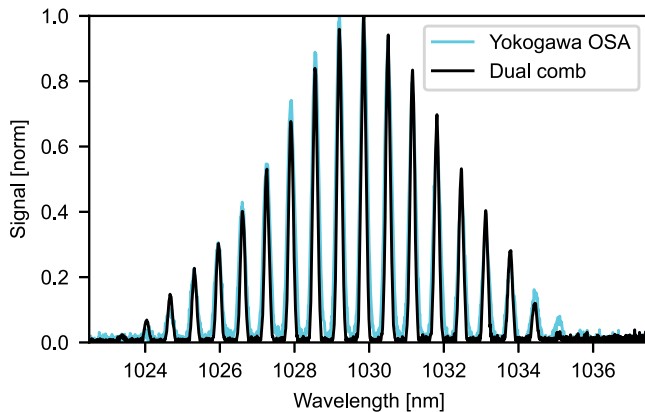

**Fig. 7 Comparison of the Fabry–Pérot etalon spectrum measured with the OSA and dual-comb setup.** The dual-comb spectrum was obtained from a single 80 μs interferogram time-trace. The width of the transmission line is 24 GHz.

spectrum are shown in Fig. 7. The retrieved spectrum was acquired from a single 80 μs interferogram time-trace. The characteristic Fabry–Pérot transmission lines are clearly visible and both measurements agree well within a reasonable deviation. Thus, the single 80 μs interferogram time-trace leads to a spectral resolution of at least 24 GHz.

Finally, to verify the real-life applicability of our dual-comb spectrometer, we performed spectroscopic measurements on acetylene ($C_2H_2$). Acetylene is a well-studied molecule having overtone absorption around 1034 nm. It has been measured in similar DCS experiments multiple times[34]. Our initial laser spectrum was long-pass filtered to have optimal overlap with the acetylene absorption spectrum around 1034 nm. Since the spectrum was narrowed, a higher $\Delta f_{rep} = 510$ Hz was realized

without aliasing due to the higher beating orders. The measurement setup is shown in more detail in the Supplementary material. With this higher $\Delta f_{rep}$, 167 interferograms were recorded within 330 ms. In the 20 μs long time trace shown in Fig. 8a, free-induction decay of acetylene can clearly be seen. The Fourier transform of this interferogram and the corresponding measurement performed by our OSA are shown in Fig. 8b. However, the dual-comb measurement takes only 20 μs while the measurement with the state-of-the-art spectrum analyzer (Yokogawa AQ6370) requires 10 min in high-resolution (0.02 nm) mode. This result confirms the applicability of our high-power dual-comb spectrometer to real-life spectroscopic measurements.

## Discussion

The passive dual-comb oscillator relies on thin-disk technology with its intrinsic power scalability. Hence, this demonstration should be able to approach the parameters of a state-of-the-art single output system of around 62 MW peak and 270 W average powers[20,27]. The presented dual-comb cavity layout is independent of the mode-locking mechanism. Therefore, an application to powerful SESAM mode-locked systems[35,36] is not out of scope. Moreover, the disk geometry, with its large, flat, and easily scalable active area, allows for a simple integration of multiple spatial modes. Thus, not only the dual-output systems but multiple-output lasers can be demonstrated. Multiple-output oscillators would open a way toward a very compact, simple, cost-effective multi-comb system (such as a tri-comb system) for 2D spectroscopy[37], and generally nonlinear multi-dimensional spectroscopy and imaging.

Interestingly, the loss of coherence between interferograms due to $\Delta f_{CEO}$ fluctuations is not evident during at least 312 ms measurement time (equivalent to 40 bursts). It was shown that a simple numerical technique to correct the interferogram arrival time can compensate for the $\Delta f_{rep}$ fluctuations sufficiently well for the presented measurements. In addition, apodization helps to increase the SNR ratio mainly by removing the side-interferograms associated with the side-pulse generated by the thin-disk dual-comb system (see Supplementary information for details). In practical dual-comb applications though, different sample types (gas, solid or liquid) and different spectral resolution can be of interest, which determine the duration of the apodization window. The ideal dual-comb system would provide delays between interferograms on the order of these response times, rendering this noise suppression scheme unnecessary. Ultimately, the spectral resolution is going to define the time window of interest. For example, in liquids (or even solids) the lines are broadened to around 100 GHz that makes it easy to record within approximately 5 μs duration of the interferogram. Gas at atmospheric pressure with about 1 GHz broadened linewidth can be

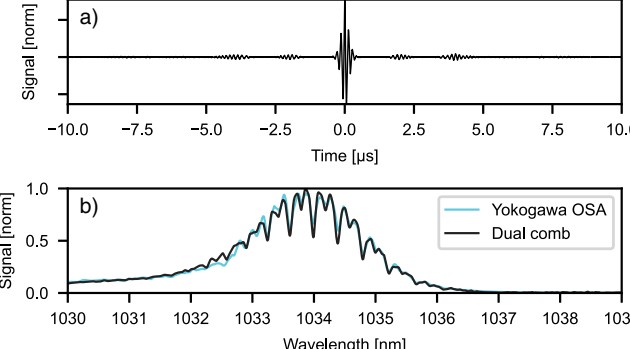

**Fig. 8 Acetylene spectroscopy. a** A 20 μs time trace with main interferogram and decay oscillations induced by acetylene is shown. **b** Spectra of a single interferogram (black) and reference spectrum (turquoise, Yokogawa OSA) are displayed.

sufficiently sampled during 0.5 ms that corresponds to 1/10th of the time window between the interferograms. Our Acetylene measurement uses a 20-μs apodization window (see Fig. 8), and reaches the 5.65 GHz resolution of Yokogawa OSA. In addition, it is necessary to state with regard to the correction of $\Delta f_{rep}$ fluctuations, that it will be more challenging to deal with interferograms having much more complex temporal structures originating from complex real-world samples.

One very attractive approach for passive $f_{CEO}$ stabilization was been demonstrated with KLM thin-disk oscillators by means of intra-pulse difference frequency generation and simultaneous broadband conversion in mid-IR range at ~100 mW average power[38]. The required spectral broadening and temporal pulse compression also were demonstrated for a large span of peak and average powers in fibers or multipass cells[39]. Furthermore, a system driven by a mode-locked thin-disk Ho:YAG oscillator running at 2.1 μm wavelength[40] showed excellent conversion efficiencies[41] and ultra-broadband coverage (500–2250 cm$^{-1}$) of the IR region. Therefore, combining the passive dual-comb approach and the Ho:YAG thin-disk technology should be possible in the near future. This would open a route toward ultra-broadband DCS of diverse chemical and biological samples[42]. Interestingly, the systems can be upgraded with an electro-optic sampling unit[6] avoiding the use of liquid nitrogen detectors.

However, the extension of the spectral range toward UV and VUV might be even more interesting as no dual-comb lasers are available in this spectral range[2]. Efficient spectral broadening or nonlinear frequency conversion to UV (257 nm) is already possible at the current power level, and it will be even simpler for power-scaled systems. The overall passive stability of the system, in particular, low $\Delta f_{CEO}$ fluctuations estimated to be 12 mrad (see Supplementary information), paves the way to UV and VUV dual-combs spectroscopy. Yet, this topic clearly requires further investigation, since the required $\Delta f_{CEO}$ relative stability scales with frequency, and harmonic generation might add significant amount of noise. The results presented provide a good basis and motivation for further research, in particular, for detailed quantitative characterization of $\Delta f_{rep}$ and $\Delta f_{CEO}$ noise. Also, the upper limit of the mutual coherence time set by $\Delta f_{CEO}$ fluctuations should be measured.

It is worth noting that relatively simple techniques can be applied to the dual-comb systems to stabilize $\Delta f_{rep}$ actively. This might be necessary once the number of interferograms becomes too large to be numerically corrected a posteriori. Relatively straightforward direct averaging of the interferograms in a field-programmable gate array can be implemented instead. Stabilizing $\Delta f_{rep}$ is expected to be simpler than stabilizing $f_{rep}$ due solely to

its small residual jitter on the order of 0.1 Hz (0.1%). The origin of $\Delta f_{rep}$ fluctuations requires further investigations. We believe that these fluctuations can be drastically reduced with acoustic isolation of the oscillator and interferometer and by optimizing the mechanical components and disk cooling flow.

A recently published dual-comb thin-disk oscillator based on polarization splitting[43] delivers similar output parameters and additionally confirms our finding with regard to the importance of $\Delta f_{rep}$ fluctuations.

We have developed a passive dual-frequency comb spectrometer based on a single free-running Yb:YAG KLM thin-disk laser. To the best of our knowledge, the peak (1.2 MW) and average power (15 W) are an order of magnitude higher than all other dual-comb systems. The two cavities emit nearly identical pulse trains, with a mutual $\Delta f_{CEO}$ coherence time of at least 2.4 s. This experimentally achieved coherence is well manifested during two single interferograms showing comb lines in the spectrum after Fourier transformation without any active stabilization or numerical correction. The passive stability of the difference in repetition rates was measured to be around 0.1 Hz over 60 s. The evaluation of the interferogram trace (see Supplementary information) shows similar stability on the ms time scale. It was shown that, in combination with the simple correction of the fluctuations of $\Delta f_{rep}$ and apodization, a narrow comb linewidth of 3.1 Hz in the RF domain has been achieved[32]. Finally, spectroscopic measurements with a Fabry–Pérot etalon and spectroscopy on acetylene ($C_2H_2$) have been demonstrated.

Ultrafast thin-disk oscillators have been demonstrated with average powers approaching 300 W and peak powers as high as 62 MW. Extension toward UV and VUV range[44,45], seems especially promising taking into account the power scalability of our platform and excellent mutual coherence of the two combs.

## Methods
The experimental setup of the laser source is explained in detail in the results section. The Fabry–Pérot etalon was built with two output couplers with 98.4% and 95% reflectivity. Further details on the acetylene measurement setup and the evaluation method are explained in the Supplementary material.

## Data availability
The unprocessed dual-comb measurement data generated during the current study are available on Zenodo: https://doi.org/10.5281/zenodo.6413816.

## Code availability
The computer codes used during the current study are described in the Supplementary information and available from the corresponding author on reasonable request.

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

## Acknowledgements
We thank Prof. F. Krausz for his insightful remarks and strong support of this research, as well as Dr. V. Pervak and his team for designing and manufacturing the optimized dispersive mirror coating. We thank Dr C. Haunhorst, D. Kiesewetter and A. Borchers for facilitating this work.

## Author contributions
K.F., J.B., K.F.M. and O.P. designed the dual-comb resonator. M.I., J.B. and K.F. realized the first proof-of-principle setup experimentally. T.H. optimized the setup and finalized the measurements. Z.C., K.F. and N.P. helped with the evaluation of the measurements. N.P. and O.P. planned the experiments and supervised the work. K.F., T.H., N.P. and O.P. wrote this manuscript.

## Funding

## Competing interests
The authors declare no competing interests.
