## [Peer Review File · Nature Communications]

Dual-comb thin-disk oscillatorREVIEWER COMMENTS

Reviewer #1 (Remarks to the Author):

The paper entitled “ Dual-comb thin disk oscillator” introduces the first dual-comb system realised with thin-disk (Yb:YAG) technology. With 0.6 MW peak power (13 W average power) this system proposed by Fritsch et al. is the most powerful dual-comb system ever reported in the literature. The implementation of a single cavity for the simultaneous generation of two combs has the important advantage to increase the coherence time between the combs themselves with respect to the usual configuration based on two separated combs: the most part of the noise is in fact common-mode noise and is rejected in the detection. The two beams inside the cavity are spatially shifted allowing the use of two different output coupler and two different end mirrors. This clever expedient has the additional benefit to allow independent adjustment of the length of the two cavities and thus to setup the repetition rate difference between the two combs in a simple way up to the kHz range. A characterisation of the system and a measurement of a Fabry Perot etalon transmission spectrum are demonstrated.

The high-power reached with this setup is of extreme importance to the extension in the UV, THz and IR/Mid-IR that are fundamental region for spectroscopic investigations.

The paper is well written and the details provided allows to reproduce the experimental setup.

The literature cited is complete and relevant to the topic. In the supplementary materials the authors present a clear and extensive comparison, summarised in a table, against the competing high-power and/or single cavity dual comb systems.

The results are novel and could be of interest for a broad audience, in particular in the field of spectroscopy.

On the basis of my previous statements I suggest the publication in Nature Communications.

I would appreciate if the authors could provide some minor revisions related to the following points:

-In the last years the field of high power ultrafast lasers is dominated by two different technologies: thin-disk lasers and fiber lasers. I think that a direct comparison between the two technologies when applied to high power dual-comb spectroscopy (and in particular single-cavity dual-comb setup), highlighting advantages and drawbacks of the two implementations could be attractive.

-The authors states that Δf_{rep} “can be easily tuned up to the kilohertz range by longitudinally translating one of the split output couplers”. Is there any reason that limits the detuning of the two repetition rate?

-The authors characterise the dual-comb system and in particular they show the improvement of the phase-corrected spectrum with respect to the non-corrected one.

I think that could be interesting to know SNRs before and after phase-correction and to have an idea of the figure of merit (Ref. [1]) of the dual-comb system.

-In Fig. 7 the measurement of the transmission of a Fabry-Perot cavity is reported. The two spectra are similar but it seems to be there is an asymmetry with respect to the center frequency: the lower frequency are in some way underestimated and the higher frequency are overestimated. Is this asymmetry ascribed to the phase-correction which introduces some artefact or there is something else related ?

Pag 1 line 23: OFC instead of OFCs and linewidth instead of lineswidth

Pag 2: There is a missing verb in the first sentence of pag. 2 “DCS is rapidly advancing...”

[1] 1. Coddington, I., Newbury, N. R. & Swann, W. Dual-comb spectroscopy. *Optica* 3, 414 (2016).

Reviewer #2 (Remarks to the Author):

The authors have presented an improved scheme of generation of a set of dual combs from a single oscillator, which has become one of the hotly pursued approaches for the realization of low-complexity dual-comb systems. The major advantage of the scheme investigated in this article is the impressively high powers of both combs directly emitted from the laser with sufficient pulse bandwidths. The limited pulse energy or peak power directly delivered from the dual-comb laser investigated so far, as summarized by the authors, poses likely limitations for many applications, and this proposed platform benefits from a free-space optics, high-power laser platform that seems to offer remedy on that aspect. While this article is an important step forward in this area, I think the authors need to address the points listed below and the paper also needs to be revised so that those claims will be better supported.

- 1) The authors named the tunability of the repetition rate difference as a major advantages over previous schemes based on birefringence and chromatic dispersion. A more accurate comparison should include those demonstrated works leveraging birefringence but offering repetition rate difference agility (Photon. Res. 6, 853 (2018); APL Photonics 4, 116102 (2019)) in contrast to Ref. [22].
- 2) It was discussed that “The authors need to further clarify the phenomenon “Surprisingly, this process also works for both lasers simultaneously.”. Does it mean that once of one of the end mirrors is perturbed, both pulses start to mode-lock? If A further investigation or discussion about the physical origin is needed.
- 3) It is less clear about how the beams are aligned at the Kerr medium, despite the fact that most part of the cavity has two separated beam paths. Are they focused at the same or different spots on that medium? Does that play an role in the ‘crosstalk’ between two combs like the phenomenon described in the previous point?
- 4) Also, the image shown in Fig. 1(c) shows some horizontal displacement between two spots while in the text the separation between the beams is described mostly in the vertical direction. Is that somewhat intentional or does that have any effect on the laser performance?
- 5) As to the phase correction algorithm used, which is also further explained in the supplement section, it is necessary to give more information about the frequency shifts one has to apply to multiple continuously acquired interferograms (Fig. 4(d) shows only 3 ones, not sure if it is always like that or following the same trend), and see if that makes sense as shifts in the difference in fceo. Without further details, it is hard to see if any role the instability in the repetition rate difference might play in such changes in the position of the recovered spectrum, which may not be best compensated by shifting the spectral center. Also, it is better to discuss the applicability of such a phase correction method in other non-static scenarios. If justified, it is also better to further explain this fast drift in Δf_{ceo} and its physical origin, as some other dual-comb lasers don't have such large and fast Δf_{ceo} changes.
- 6) For the result shown in Fig. 6, which has a narrow line width, are the interferogram peaks are windowed and tails filled with zeros as described in Supplement? If so, does this ‘clean-up’ of the original data give better than normal results in that figure?
- 7) Also, the authors noted that the method used in the noise reduction described in the supplement section is not applicable to results with samples, then it sounds somewhat not very useful for practical dual-comb applications that always deal with certain samples with complex spectral response.
- 8) On page 5, in line 91, it is said that “they experience nearly equal perturbations originating from air flows, mechanical vibrations and the intensity noise of the pump source.”. Yet seems that for some vibrations such as tilt or wobbling of the mount. it would affect the combs differently. So the above statement needs to be revised.
- 9) The repetition rate difference is said to be tunable, but it is better to show the range or limits of the repetition rate difference tunability with sufficient data to support the claim.
- 10) In figure 4(a), the interferogram trace seems to show some relatively large one-to-one amplitude fluctuations. Can the authors comment further on the cause of this and further quantify this fluctuation?

11) Can the authors further explain how the mutual coherence time between the two frequency combs is estimated to be around 1ms? Is there any additional experimental result to support this claim?

Reviewer #3 (Remarks to the Author):

Fritsch et al. present a laser oscillator that is capable of “dual comb operation” at elevated average powers up to 12 watts. A very basic preliminary proof of principle experiment with a Fabry Perot etalon is demonstrated. The main selling point of the work is the potential expansion of dual comb spectroscopy into the ultraviolet region. However, no efforts have been taken to demonstrate it. With this the manuscript remains extremely superficial and is a paper interesting for laser engineers only. The high standards of Nature Communications are not met nowhere near enough. For this for example, a specific science case in the UV would have been needed to be identified that could be resolved for the first time (e.g. due to the higher spectral resolution). Also, a serious discussion on the frequency comb operation is missing (stabilizability of comb parameters, noise performance). For those reasons, I recommend to reject the manuscript from publication in Nature Communications.

Dear Reviewers,

Thank you very much for all your feedback you have provided to our manuscript. Unluckily, the submission of this manuscript has coincided with the relocation of the research group of Oleg Pronin from Munich to Hamburg. Setting up the new labs, hiring new PhD students and moving some equipment (such as the dual-comb oscillator) were the main priorities during the last one and a half years. Finally, we were able to recover the dual-comb oscillator. We also realized that this oscillator has to be properly engineered and essentially rebuilt to provide the long-term stable and reproducible measurements, so important to perform the deep-UV dual-comb spectroscopy in the future. While we were trying to improve the system and run the oscillator under low pressure/vacuum an accident happened. Namely, the laser cover has crashed, destroyed a lot of optics and caused a water shower inside the oscillator. As for now the oscillator has been nearly fully recovered from this damage.

Importantly, thanks to your comments, we realized that the fluctuations in Δf_{rep} (and not Δf_{ceo}) is the main noise source and have to be numerically corrected. In contrast to Δf_{rep} fluctuations, Δf_{ceo} fluctuations have negligible influence on the spectral behaviour.

We have tried our best to answer and address the comments of reviewers. Some of those comments will be answered much better once we are able to perform systematic and long-term measurements with the recovered from damage and re-engineered dual-comb oscillator. Meanwhile, a research group of Prof. Suedmeyer has joined the dual-comb thin-disk oscillator development (*N. Madsching, et.al., "High-power dual-comb thin-disk laser oscillator for fast high-resolution spectroscopy," Opt. Express, OE 29, 15104 (2021)*). They fairly cite our work available on ArXiv as a prior art and first thin-disk laser system. This is an additional reason why we keep this claim unchanged in this revised version of the manuscript. Their laser system based on polarization splitting delivers pretty similar output parameters and additionally confirms our funding with regard to the importance of Δf_{rep} fluctuations.

Thank you in advance for giving us a chance to share these results with the community in Nature Communications.

Reviewer #1 (Remarks to the Author):

The paper entitled “ Dual-comb thin disk oscillator” introduces the first dual-comb system realised with thin-disk (Yb:YAG) technology. With 0.6 MW peak power (13 W average power) this system proposed by Fritsch et al. is the most powerful dual-comb system ever reported in the literature. The implementation of a single cavity for the simultaneous generation of two combs has the important advantage to increase the coherence time between the combs themselves with respect to the usual configuration based on two separated combs: the most part of the noise is in fact common-mode noise and is rejected in the detection. The two beams inside the cavity are spatially shifted allowing the use of two different output coupler and two different end mirrors. This clever expedient has the additional benefit to allow independent adjustment of the length of the two cavities and thus to setup the repetition rate difference between the two combs in a simple way up to the kHz range. A characterisation of the system and a measurement of a Fabry Perot etalon transmission spectrum are demonstrated.

The high-power reached with this setup is of extreme importance to the extension in the UV, THz and IR/Mid-IR that are fundamental region for spectroscopic investigations.

The paper is well written and the details provided allows to reproduce the experimental setup. The literature cited is complete and relevant to the topic. In the supplementary materials the authors present a clear and extensive comparison, summarised in a table, against the competing high-power and/or single cavity dual comb systems.

The results are novel and could be of interest for a broad audience, in particular in the field of spectroscopy.

On the basis of my previous statements I suggest the publication in Nature Communications. I would appreciate if the authors could provide some minor revisions related to the following points:

-In the last years the field of high power ultrafast lasers is dominated by two different technologies: thin-disk lasers and fiber lasers. I think that a direct comparison between the two technologies when applied to high power dual-comb spectroscopy (and in particular single-cavity dual-comb setup), highlighting advantages and drawbacks of the two implementations could be attractive.

Thank you for this comment. Indeed, it is quite a complex task to provide an objective and fair comparison between fiber-based and thin-disk technology. Such a comparison should cover a broad parameter range including emission bandwidth, pulse & beam quality, polarization state, noise behaviour as well as achievable peak and average powers (please see such a comparison here https://www.rp-photonics.com/thin_disk_lasers.html). We have also cited this article of Paschotte in our manuscript: “Further reading and references on the comparison of the thin-disk laser technology and other approaches is summarized in [18].” Since the choice of these laser parameters strongly depends on the experiment and application, it is extremely hard to provide an objective comparison in this manuscript and still keep it within the 5000 words limit.

-The authors states that Δf_{rep} "can be easily tuned up to the kilohertz range by longitudinally translating one of the split output couplers". Is there any reason that limits the detuning of the two repetition rate?

Initially we were interested in the detuning range 100 Hz - 1kHz particularly relevant for the dual-comb spectroscopy. We have experimentally verified that the dual comb system can easily be tuned within this range and can be detuned even up to 1 MHz. Practically there is no limit for the lower detuning value. However, there can be a limit for the upper detuning value. There are two possible reasons for it. 1. The detuning of one of the cavities changes the roundtrip-time and, thus the pulse train repetition rate. This, in turn, influences the pulse energy $E_p = f_{\text{rep}} \cdot P_{\text{avg}}$. Since all optical elements are shared in the laser cavities (Kerr medium, separation between the focusing curved mirrors), Kerr-lens mode-locking is optimized for only a certain pulse energies range. Considering our previous experiments, we can estimate that 5% energy decrease (or increase) will lead to the instabilities in one of the arms. This 5% change in the cavity length corresponds to approximately 3 MHz maximal acceptable detuning. 2. The detuning of one of the cavity arms changes the stability zone of this cavity. As Kerr-lens mode-locking is quite sensitive to the cavity position in the stability zone, this will manifest itself in a decreased pulse energy of the detuned cavity up to the point where this cavity will cease mode-locking. These limits depend strongly on the specific cavity configuration. We have implemented this comment in the supplementary material. Also we have commented on it in the manuscript "The tuning up to 1 MHz was experimentally verified".

-The authors characterise the dual-comb system and in particular they show the improvement of the phase-corrected spectrum with respect to the non-corrected one. I think that could be interesting to know SNRs before and after phase-correction and to have an idea of the figure of merit (Ref. [1]) of the dual-comb system.

Thank you very much for this instructive comment which motivated us to look more carefully in the data analysis we have performed. Firstly, it turned out that the fluctuations in Δf_{rep} (and not Δf_{ceo}) had to be corrected. In contrast to Δf_{rep} fluctuations, Δf_{ceo} fluctuations have negligible influence on the spectral behaviour. Secondly, apodization plays a crucial role in increasing SNR. We have specified the SNR values directly in the manuscript as follows: "A comparison between the spectra with the apodization and the spectra without any correction shows that the SNR=486 for the apodized case is significantly improved compared to the spectra without any correction with SNR=18.

Fig. 5b depicts optical spectra with the already mentioned Δf_{rep} correction technique applied including the apodization and the spectra without any correction. This case shows SNR=1454 compared to SNR=18 of the non-corrected case while the good agreement with the reference measurement remains unchanged".

We estimate the figure of merit to be approximately $4 \cdot 10^5$ according to Ref. [1].

-In Fig. 7 the measurement of the transmission of a Fabry-Perot cavity is reported. The two spectra are similar but it seems to be there is an asymmetry with respect to the center frequency: the lower frequency are in some way underestimated and the higher frequency are overestimated. Is this asymmetry ascribed to the phase-correction which introduces some artefact or there is something else related ?

Thank you for this comment, indeed, we also noticed this asymmetry when performing the measurements. Unfortunately, we had no time to investigate these observations in more detail. Currently we are reengineering and optimizing the dual comb system. Afterwards we plan to characterize the intensity fluctuations of both combs independently and check whether the noise contributions are identical for both of them. Additionally, more quantitative measurements are planned to characterize the relative fceo jitter and perform real-time Acetylene measurements. All these reasons can contribute to the asymmetry observed and would be even more critical for UV and deep UV regions.

Pag 1 line 23: OFC instead of OFCs and linewidth instead of lineswidth

Pag 2: There is a missing verb in the first sentence of pag. 2 “DCS is rapidly advancing...”

Thank you very much for pointing out these blunders. They were corrected in the manuscript.

[1] 1. Coddington, I., Newbury, N. R. & Swann, W. Dual-comb spectroscopy. *Optica* 3, 414 (2016).

Reviewer #2 (Remarks to the Author):

The authors have presented an improved scheme of generation of a set of dual combs from a single oscillator, which has become one of the hotly pursued approaches for the realization of low-complexity dual-comb systems. The major advantage of the scheme investigated in this article is the impressively high powers of both combs directly emitted from the laser with sufficient pulse bandwidths. The limited pulse energy or peak power directly delivered from the dual-comb laser investigated so far, as summarized by the authors, poses likely limitations for many applications, and this proposed platform benefits from a free-space optics, high-power laser platform that seems to offer remedy on that aspect. While this article is an important step forward in this area, I think the authors need to address the points listed below and the paper also needs to be revised so that those claims will be better supported.

1) The authors named the tunability of the repetition rate difference as a major advantages over previous schemes based on birefringence and chromatic dispersion. A more accurate comparison should include those demonstrated works leveraging birefringence but offering repetition rate difference agility (*Photon. Res.* 6, 853 (2018); *APL Photonics* 4, 116102 (2019)) in contrast to Ref. [22].

Thank you very much for pointing it out. The mentioned work was published just four months prior to our submission and we were actually unaware of it. This publication is of importance in the context of our work, since it pushes the viable repetition rate tuning range for birefringence based dual-comb fiber systems. We modified our manuscript accordingly: “With the exception for work [24,25], those systems are quite limited in the tunability and tunability range (see supplementary table 1 for more information),....” Also, we added these references in our system comparison in the supplementary material.

2) It was discussed that “The authors need to further clarify the phenomenon “Surprisingly, this process also works for both lasers simultaneously.”. Does it mean that once of one of the end mirrors is perturbed, both pulses start to mode-lock? If A further investigation or discussion about the physical origin is needed.

You are right, our sentence might be understood that the mode-locking one of the cavities will also induce pulsed operation in the other one. However, this is explicitly not the case. There is no mechanism which couples the cavities in such a way. We modified the manuscript accordingly to highlight this fact more clearly. "...both cavities can be mode-locked routinely at the same time by manually perturbing one of the telescope mirrors."

3) It is less clear about how the beams are aligned at the Kerr medium, despite the fact that most part of the cavity has two separated beam paths. Are they focused at the same or different spots on that medium? Does that play an role in the 'crosstalk' between two combs like the phenomenon described in the previous point?

The beams are not perfectly parallel before and after the focusing telescope, i.e. the beams are focused to two distinct, vertically separated, spots in the Kerr-medium. The offset is large enough to be visible by eye and an infrared-viewer. This implies no cross-talk between the modes in the Kerr-medium. We modified the manuscript to clarify this "A telescope comprising two concave mirrors with a radius of curvature (ROC) of 300 mm focuses both cavity modes into two distinct foci within a Kerr medium composed of a 3 mm sapphire plate placed under Brewster's angle."

4) Also, the image shown in Fig. 1(c) shows some horizontal displacement between two spots while in the text the separation between the beams is described mostly in the vertical direction. Is that somewhat intentional or does that have any effect on the laser performance?

This small offset (less than the radius of the depletion zone) has two factors contributing to it. Firstly, we use a pilot laser to pre-align both cavities before starting the actual laser operation. Here we try to avoid the horizontal offset, but there is a small mismatch between the pilot laser and the final laser mode. And secondly, once both laser cavities are running in the mode-locked operation, we optimize the laser alignment to achieve maximum output power in both arms. It turns out that this optimized alignment yields the highest output powers. The comment was implemented in Fig.1 capture "They show a slight horizontal displacement which is a result of the optimized alignment."

5) As to the phase correction algorithm used, which is also further explained in the supplement section, it is necessary to give more information about the frequency shifts one has to apply to multiple continuously acquired interferograms (Fig. 4(d) shows only 3 ones, not sure if it is always like that or following the same trend), and see if that makes sense as shifts in the difference in f_{ceo} .

Thank you very much for this instructive comment which motivated us to look more carefully in the data analysis we have performed. Firstly, it turned out that the fluctuations in Δf_{rep} (and not Δf_{ceo}) had to be corrected. In contrast to Δf_{rep} fluctuations, Δf_{ceo} fluctuations have negligible influence on the spectral behaviour. We have described the compensation of Δf_{rep} fluctuations in detail in the supplementary information. Additionally we have analyzed approximately 200 interferograms from the trace. The standard deviation of the central frequency fluctuations corresponds to about 0.8 Hz. This value is directly proportional to Δf_{ceo} fluctuations and is, thus, equal to 24 mrad RMS. More details are provided in the supplementary chapter.

Without further details, it is hard to see if any role the instability in the repetition rate difference might play in such changes in the position of the recovered spectrum, which may not be best compensated by shifting the spectral center.

Exactly, this instability is the dominant one. As for now we don't apply any phase compensation due to its high passive stability and do only correction of Δf_{rep} fluctuations. Additionally, it turns out that the correction of Δf_{rep} fluctuations does not influence the central frequency fluctuations of the recovered spectrum.

Also, it is better to discuss the applicability of such a phase correction method in other non-static scenarios.

We combine the answer to this question with the answer to your question 7).

If justified, it is also better to further explain this fast drift in Δf_{ceo} and its physical origin, as some other dual-comb lasers don't have such large and fast Δf_{ceo} changes.

Actually, as it turns out, the Δf_{ceo} fluctuations are rather negligible for now.

The Δf_{rep} fluctuations are not so fast, they are happening on the time scale 10-100 ms (see Fig. 1 supplementary).

6) For the result shown in Fig. 6, which has a narrow line width, are the interferogram peaks are windowed and tails filled with zeros as described in Supplement? If so, does this 'clean-up' of the original data give better than normal results in that figure?

This clean-up or apodization has significant influence on the background noise suppression as shown in Fig.5 of the manuscript. This step is mainly necessary to eliminate the side interferograms generated in the trace. Unfortunately, for now we cannot suppress the side pulse causing these side interferograms.

7) Also, the authors noted that the method used in the noise reduction described in the supplement section is not applicable to results with samples, then it sounds somewhat not very useful for practical dual-comb applications that always deal with certain samples with complex spectral response.

This is indeed not very clear in the manuscript. In practical dual-comb applications though, different sample types (gas, solid or liquid) and different spectral resolution can be of interest, which determine the duration of this "window". The ideal dual-comb system would provide delays between interferograms on the order of these response times, rendering this noise suppression scheme unnecessary. Ultimately, the spectral resolution is going to define the time window of interest. For example, in liquids (or even solids) the lines are broadened to around 100 GHz which makes it easy to record within approximately 5 μs duration of the interferogram. Gas at atmospheric pressure with about 1 GHz broadened linewidth can be sufficiently sampled during 0.5 ms which corresponds to 1/10th of the time window between the interferograms.

We have done a few preliminary measurements of Acetylene under 2 bar pressure and see that the lines can be well resolved with the apodization technique.

8) On page 5, in line 91, it is said that “they experience nearly equal perturbations originating from air flows, mechanical vibrations and the intensity noise of the pump source.”. Yet seems that for some vibrations such as tilt or wobbling of the mount. it would affect the combs differently. So the above statement needs to be revised.

We have revised the sentence in the manuscript in the following way: “They experience nearly equal perturbations originating from air flows, mechanical vibrations and the intensity noise of the pump source. However, the system can be somewhat more sensitive to the drifts and vibrations associated with the displacement of the beams in a plane parallel to the optical table”

9) The repetition rate difference is said to be tunable, but it is better to show the range or limits of the repetition rate difference tunability with sufficient data to support the claim.

We have implemented a sentence in the manuscript ““The tuning up to 1 MHz was experimentally verified” and additionally commented on it in the supplementary information.

10) In figure 4(a), the interferogram trace seems to show some relatively large one-to-one amplitude fluctuations. Can the authors comment further on the cause of this and further quantify this fluctuation?

There are two major contributions to these amplitude fluctuations. Firstly, the amplitude noise of our thin-disk oscillator contributes to it. We have extensively characterised and minimized this noise in our previous work [41,42, dissertation of M. Seidel <https://edoc.ub.uni-muenchen.de/23640/>] and plan to perform similar work for the well engineered version of the dual-comb system which is currently under development. Taking into account our experience in the development of the Kerr-lens mode-locked thin-disk oscillators, it should be possible to reach intensity fluctuations of <0.3% RMS for the optimized dual-comb system. Secondly, the sampling of the interferograms is just about sufficiently dense, but not oversampled. This means that the height of each peak of a single interferogram is not recorded accurately for each interferogram and therefore the absolute values cannot be directly compared. We have improved it just a few weeks ago and could make a few fresh measurements before the laser system was “damaged”.

11) Can the authors further explain how the mutual coherence time between the two frequency combs is estimated to be around 1ms? Is there any additional experimental result to support this claim?

We have slightly changed this claim. “Both cavities emit nearly identical pulse trains, with the mutual Δf_{ceo} coherence time of approximately 2.4 s”. On the other hand, taking into account Figure 1 demonstrating Δf_{rep} fluctuations analysis, the jitter between neighboring interferograms separated by ca. 5 ms is negligible which indicates the coherence time somewhat on the order of 5 ms.

Reviewer #3 (Remarks to the Author):

Fritsch et al. present a laser oscillator that is capable of “dual comb operation” at elevated average powers up to 12 watts. A very basic preliminary proof of principle experiment with a Fabry Perot etalon is demonstrated. The main selling point of the work is the potential expansion of dual comb spectroscopy into the ultraviolet region. However, no efforts have

been taken to demonstrate it. With this the manuscript remains extremely superficial and is a paper interesting for laser engineers only. The high standards of Nature Communications are not met nowhere near enough. For this for example, a specific science case in the UV would have been needed to be identified that could be resolved for the first time (e.g. due to the higher spectral resolution). Also, a serious discussion on the frequency comb operation is missing (stabilizability of comb parameters, noise performance). For those reasons, I recommend to reject the manuscript from publication in Nature Communications.

We thank the reviewer for providing this opinion on our manuscript.

However, we respectfully disagree with this opinion and provide our scientific argumentation below.

Our article reports on an original laser oscillator design, that enables for the first time high average power and peak power for a dual-comb system. This design was made possible thanks to creative thinking combined with extensive and specialized knowledge of the physics of thin-disk lasers. This goes well beyond the engineering capabilities and skills. The objectives of high pulse energy and high average power for dual-comb interferometry have been long sought after. Our work shows a solution to this issue that improves the state-of-the-art by over one-order of magnitude. We therefore believe that our paper will be of interest:

- i) to laser physicists interested in the laser design and new applications of thin-disk oscillators
- ii) to spectroscopists interested in the new tool for experiments in nonlinear dual-comb spectroscopy (where nonlinear effects are generated at the sample) or in dual-comb spectroscopy in spectral regions that are only accessible through nonlinear frequency conversion of the laser light.
- iii) to other developers and users of dual-comb interferometers, interested in, for instance, optical coherence tomography or laser ranging.

We point out, as a specific example, that there is a particular need for high-power dual-comb systems to access the ultraviolet range. With the one-order of magnitude improvement in power of our system, we do believe that our work is a significant milestone towards such a feat. We also mention (page 10) that many other obstacles, related to the higher requirements in relative stability and in phase noise multiplication by harmonic conversion need to be overcome.

The development of high-resolution spectroscopy instruments is a long-term endeavor, especially in the ultraviolet region.

To support our statements, we find the example of the history of instrumentation in Michelson-based Fourier transform spectroscopy particularly instructive. A simplified chronology is the following:

- 1891: A.A. Michelson points out that his interferometer could be suitable for spectroscopy analysis (A.A. Michelson, On the application of interference-methods to spectroscopic measurements, *Philosophical Magazine* 31, 338-346 (1891).
- 1951 : In this thesis, P. Fellgett identifies the multiplex advantage §P. Fellgett, Thesis, Univ. of Cambridge, 1951).
- 1954: P. Jacquinot points out the throughput advantage of interferential spectrometers (P. Jacquinot, 17e Congrès du GAMS, Paris, p. 25 (1954)).

- 1961: The thesis of J. Connes, with its in-depth analysis of the potential of the technique, establishes the fundamental principles of Fourier transform spectroscopy (J. Connes, Spectroscopic Studies Using Fourier Transformations, Rev. Opt. 40, 45-260 (1961).)
- 1965: J.W. Cooley and J. Tukey popularize the Fast Fourier transform algorithm.
- 1970: The first mid-infrared spectra demonstrating all the advantages of Fourier transform spectroscopy in terms of resolution, recording time, sensitivity, precision, and spectral span are published by the group of P. Connes (J. Connes et al. Nouvelle Revue d'Optique Appliquée 1, 3-22 (1970)).
- 1978: The first near-ultraviolet (360 nm) high-resolution Fourier transform spectra are demonstrated (P. Luc et al. Appl. Opt. 17, 1327-1331 (1978)).
- 2011: The first vacuum-ultraviolet high-resolution Fourier transform spectra are published (N. de Oliveira, Nature Photon 5, 149–153 (2011)).

The need for better spectrometers in the ultraviolet region is widely documented in the literature. As this manuscript paves a way this goal, we do not really see the need for discussing possible projects at this stage. We have nevertheless added to our manuscript the following references, which provide some general argumentation and various examples relevant to different fields of science:

- Environmental sciences - field measurements and laboratory reference data
J. Orphal et al., Absorption cross-sections of ozone in the ultraviolet and visible spectral regions: Status report 2015, Journal of Molecular Spectroscopy 327, 105-121 (2016).
- Astrophysics and astronomy – laboratory reference data
M.T. Belmonte et al., The laboratory astrophysics spectroscopy programme at Imperial College London, Galaxies 6, 109 (2018)
- Precision molecular spectroscopy for tests of fundamental physics
W. Ubachs et al., Physics beyond the Standard Model from hydrogen spectroscopy, Journal of Molecular Spectroscopy 320, 1-12 (2016).
- Precision atomic spectroscopy for tests of fundamental physics
A. Cingöz, et al. Direct frequency comb spectroscopy in the extreme ultraviolet. Nature 482, 68–71 (2012).
- Development of the next generation of nuclear clocks
Benedict Seiferle, et al., Energy of the ^{229}Th nuclear clock transition, Nature 573, 243–246(2019)

Additionally, as to us, the main interest in dual-comb spectroscopy is, however, not the high resolution, which can be similar or even higher with other types of laser systems. The main interest of dual-comb spectroscopy is the overall consistency of the spectra, which is granted by the unique combination of multiplex recording and absolute-frequency-calibration by the combs.

REVIEWER COMMENTS

Reviewer #2 (Remarks to the Author):

The authors have made some changes to the manuscript after a relatively long period of time (during which there was a demonstration of a similar scheme), due to, as claimed, logistic and experimental incidents. In their response and revision, they addressed several of my previous questions. They have taken the time to also revise some technical details such as how the data are processed and get better or more accurate understanding of the impact of repetition rate difference on their measurements.

There are still a couple of important points that I feel need to be further clarified or improved, though:

1) I'm afraid that the results shown in the manuscript about the SNR and spectrum after 'apodization' are, even if technically seemingly correct, kind of misleading, especially when used to compare with the more widely accepted standard approach, such as that summarized in Ref. 12. The huge increase in SNR and cleaning up of the spectrum is due to the very narrow window applied during 'apodization'. The explanation of some rather periodic 'noise' in the temporal trace is kind of vague. As a temporal measurement scheme, the 'windowing' certainly would result in such an 'improvement'. However, when compared to other systems not doing that, such a parameter (SNR) is no longer a fair one, as it highly depends on the window size. It no longer represents the characteristics of the physical system people want to know. It also hugely limits the spectral resolution, which in this demonstration is not required to be high. While the authors argued that many scenarios don't demand very high resolution, as a technique or laser source that is to be applied to different future systems, giving the correct and fair parameter to show its capability (or shortcomings) is quite important.

2) There are a few places that the authors claimed that they were to make improvements or had made some different measurements to justify their claims, without showing additional results. I would be better to see some of them, instead of just 'trust' what had been said blindly.

3) The answer to my previous question 10) says "the sampling of the interferograms is just about sufficiently dense, but not oversampled. This means that the height of each peak of a single interferogram is not recorded accurately for each interferogram" What is the sampling rate? Why can't it be faster? It's often not a limiting factor in such an experiment.

4) There are still quite some typos or grammar errors, I think, such as "continious" in the supplement.

Reviewer #3 (Remarks to the Author):

The revised manuscript "Dual comb thin disk oscillator" by the authors Fritsch et al. has been re-submitted after more than one year. The authors comprehensibly explain this long period with the relocation of the group to a new university and several setbacks with the experimental (laser) setup.

The authors still could and should have used those months to implement some of the reviewers' comments more thoroughly and to update the reference list that became in the meantime partially outdated. Because the two other reviewers are in favor of publishing this manuscript in Nature Communications, I would agree with publishing in this journal after eliminating the following three main open weaknesses of the revised manuscript:

Point 1

Reviewer 1 made the great suggestion to directly compare the two technologies of thin disk and fiber based (DCS) systems. This should be included with a table stating the main parameters important for DCS in the supplementary file because it will make the overall significance of this work much more accountable. Only referencing the Paschotta link is too superficial as it gives only an introducing overview to thin disk lasers but does not account at all for the dual comb applications.

Point 2

The authors give reasonable answers to the questions # 5, 6 and 7 of reviewer 2 in the answering letter but neither in the manuscript nor in the supplementary file. Those answers should be addressed shortly in the manuscript (or SI) files also for the general reading audience because any reader could come up with similar thoughts as the reviewer. Short notes/1-2 sentences max. should be sufficient.

Point 3

The cited references are in the meantime not up to date - displaying the "most recent" reference from 2019. This list has to be updated. A quick literature check brings novel publications realizing single laser DCS systems like

2020: <https://doi.org/10.1364/OE.403072>,
https://doi.org/10.1364/CLEO_SI.2020.SF3H.2

The potential applications in the UV should be emphasized with the following works

2020: <http://dx.doi.org/10.3390/rs12203444>
2021: <https://doi.org/10.1364/OE.424940>

Maybe there is even more recent work to be cited, please check.

Dear Reviewers,

Thank you very much for your comments. We have tried our best to implement those.

We submit our manuscript with the “tracked” changes such that it is easier for you to read and identify the changes. Additionally, new sentences are highlighted in red.

Reviewer #2 (Remarks to the Author):

The authors have made some changes to the manuscript after a relatively long period of time (during which there was a demonstration of a similar scheme), due to, as claimed, logistic and experimental incidents. In their response and revision, they addressed several of my previous questions. They have taken the time to also revise some technical details such as how the data are processed and get better or more accurate understanding of the impact of repetition rate difference on their measurements.

Thank you for your positive feedback here.

There are still a couple of important points that I feel need to be further clarified or improved, though:

1) I'm afraid that the results shown in the manuscript about the SNR and spectrum after 'apodization' are, even if technically seemingly correct, kind of misleading, especially when used to compare with the more widely accepted standard approach, such as that summarized in Ref. 12.

It is important and crucial to avoid any misleading specifications in our manuscript. So, every time in our manuscript we mention SNR value we clearly state that this SNR is performed with the apodization of the interferogram. For example, we have implemented the following abbreviations SNR, \$SNR_{\text{time-corr.}}\$, \$SNR_{\text{time-corr., apod.}}\$ to avoid any misunderstandings. These SNR values can be now clearly identified by readers and used as fair parameters for the comparison with other systems.

The huge increase in SNR and cleaning up of the spectrum is due to the very narrow window applied during 'apodization'.

We have even added this sentence directly in the manuscript to clearly and transparently state the effect of apodization.

“The huge increase in SNR and cleaning up of the spectrum is due to the relatively narrow window applied during 'apodization'”

The explanation of some rather periodic 'noise' in the temporal trace is kind of vague. As a temporal measurement scheme, the 'windowing' certainly would result in such an 'improvement'.

We would like to emphasise here that the “windowing” or apodization is not our invention. The apodization is a well known and frequently applied technique in Fourier Transform Spectroscopy. For example, the book sub-chapter 2.4 from Peter R. Griffiths “Fourier transform infrared spectroscopy” is fully devoted to this technique. Conceptually dual-comb spectroscopy is similar to Fourier Transform Spectroscopy and we transfer that knowledge

from Fourier Transform Spectroscopy to dual-comb spectroscopy. The apodization is an important part of this knowledge transfer.

However, when compared to other systems not doing that, such a parameter (SNR) is no longer a fair one, as it highly depends on the window size. It no longer represents the characteristics of the physical system people want to know. It also hugely limits the spectral resolution, which in this demonstration is not required to be high. While the authors argued that many scenarios don't demand very high resolution, as a technique or laser source that is to be applied to different future systems, giving the correct and fair parameter to show its capability (or shortcomings) is quite important.

As addressed above we introduced different SNR parameters (SNR , $SNR_{\text{time-correct}}$, $SNR_{\text{time-corr., apod}}$) to avoid any misunderstandings and keep things transparent.

Additionally, we have performed Acetylene measurements and compared them to the measurements with a commercial state of the art optical spectrum analyzer from Yokogawa. The agreement between the measurements (see Fig.8) shows that our passive dual-comb spectrometer based on thin-disk technology works at least as good as our Yokogawa AQ6370D.

2) There are a few places that the authors claimed that they were to make improvements or had made some different measurements to justify their claims, without showing additional results. I would be better to see some of them, instead of just 'trust' what had been said blindly.

We carefully went through our answers and tried to identify those promises/claims we made. They are highlighted in yellow:

a) We also realized that this oscillator has to be properly engineered and essentially rebuilt to provide the long-term stable and reproducible measurements.

Indeed, we decided that rebuilding the current system would strongly slow us down in dual-comb spectroscopy and, especially, in moving towards dual-comb system for UV spectral range. So, we decided to build one more thin-disk dual-comb oscillator with higher average and peak power in parallel from scratch. As a proof you can see this photo.

In this photograph you can see proper steel flexure mirror mounts with low-thermal expansion design and excellent Polaris mounts from Thorlabs. This housing is designed to be evacuated to prevent air fluctuations and show no mechanical deformations during the evacuation. Essentially, this is the next-generation system and logical evolution step towards long-term dual-comb measurements.

- b) Thank you for this comment, indeed, we also noticed this asymmetry when performing the measurements. Unfortunately, we had no time to investigate these observations in more detail. Currently we are reengineering and optimizing the dual comb system. Afterwards we plan to characterize the intensity fluctuations of both combs independently and check whether the noise contributions are identical for both of them. Additionally, more quantitative measurements are planned to characterize the relative fceo jitter and perform real-time Acetylene measurements.

As mentioned above the completely new dual-comb thin-disk oscillator is under development. So, this is a work in progress. Additionally, we have already optimized the current laser system:

- we increased the peak power by factor 2
- we decreased pulse duration by 40%
- we implemented new measurements with the spectral span as large as 40 nm, before it was ~4 nm
- we managed through careful dispersion management to strongly suppress side-pulses and this way avoid satellite interferograms

All these changes are clearly reflected in the manuscript.

The intensity fluctuations strongly depend on the position at the resonator stability edge and are extremely sensitive to the alignment. As for now, any optimization of those fluctuations would require the re-engineered system, which is under development.

- c) This clean-up or apodization has significant influence on the background noise suppression as shown in Fig.5 of the manuscript. This step is mainly necessary to eliminate the side interferograms generated in the trace. Unfortunately, for now we cannot suppress the side pulse causing these side interferograms.

The laser system upgrade and dispersion management, indeed, improved the situation with the side-pulses. They are not anymore that strongly pronounced. We still see them in the autocorrelation measurements (see supplement), however, we cannot resolve them in Fig. 8a showing the Acetylene interferogram.

- d) We have done a few preliminary measurements of Acetylene under 2 bar pressure and see that the lines can be well resolved with the apodization technique.

These measurements are now implemented in the manuscript.

- e) We have extensively characterised and minimized this noise in our previous work [41,42, dissertation of M. Seidel <https://edoc.ub.uni-muenchen.de/23640/>] and plan to

perform similar work for the well engineered version of the dual-comb system which is currently under development. Taking into account our experience in the development of the Kerr-lens mode-locked thin-disk oscillators, it should be possible to reach intensity fluctuations of <0.3% RMS for the optimized dual-comb system. Secondly, the sampling of the interferograms is just about sufficiently dense, but not oversampled. This means that the height of each peak of a single interferogram is not recorded accurately for each interferogram and therefore the absolute values cannot be directly compared. We have improved it just a few weeks ago and could make a few fresh measurements before the laser system was "damaged".

As promised, new measurements were recorded, as you have suggested, at a higher sampling rate of 625 MSa/s, as compared to 250 Msa/s in the previous measurement.

As mentioned above, the completely new thin-disk dual-comb system is under development.

3) The answer to my previous question 10) says "the sampling of the interferograms is just about sufficiently dense, but not oversampled. This means that the height of each peak of a single interferogram is not recorded accurately for each interferogram" What is the sampling rate? Why can't it be faster? It's often not a limiting factor in such an experiment.

For the previous measurements we used a DAQ card with max. 250 MSa/s sampling rate. The minimum sampling rate required for our laser system is the Nyquist limit of $2 \times 60 \text{ MHz} = 120 \text{ MSa/s}$, so 250 MSa/s would be "just about sufficiently dense". The measurements presented in the updated manuscript were recorded with a new oscilloscope at 625 MSa/s. Further increase of the sampling rate is rather impractical due to the limited memory capacity of the oscilloscope and, thus, limited duration of the time trace.

4) There are still quite some typos or grammar errors, I think, such as "continious" in the supplement.

The manuscript and supplementary information were sent to a professional native-English proof-reading service. Thus, this issue should be completely fixed.

Reviewer #3 (Remarks to the Author):

The revised manuscript "Dual comb thin disk oscillator" by the authors Fritsch et al. has been re-submitted after more than one year. The authors comprehensibly explain this long period with the relocation of the group to a new university and several setbacks with the experimental (laser) setup.

The authors still could and should have used those months to implement some of the reviewers' comments more thoroughly and to update the reference list that became in the meantime partially outdated. Because the two other reviewers are in favor of publishing this

manuscript in Nature Communications, I would agree with publishing in this journal after eliminating the following three main open weaknesses of the revised manuscript:

The manuscript has been updated with the new measurements, so we really tried to make use of that time to provide new insights and better measurements in the manuscript.

Point 1

Reviewer 1 made the great suggestion to directly compare the two technologies of thin disk and fiber based (DCS) systems. This should be included with a table stating the main parameters important for DCS in the supplementary file because it will make the overall significance of this work much more accountable. Only referencing the Paschotta link is too superficial as it gives only an introducing overview to thin disk lasers but does not account at all for the dual comb applications.

A new table comparing the two technologies has been introduced into the supplementary information.

Point 2

The authors give reasonable answers to the questions # 5, 6 and 7 of reviewer 2 in the answering letter but neither in the manuscript nor in the supplementary file. Those answers should be addressed shortly in the manuscript (or SI) files also for the general reading audience because any reader could come up with similar thoughts as the reviewer. Short notes/1-2 sentences max. should be sufficient.

#5 from Reviewer 2 has been addressed directly in the manuscript

Careful analysis of the DCS time trace (see Fig. 4a) shows that Δf_{rep} fluctuations (see supplement 1 for details) are non-negligible, and it confirms the standard deviation of approximately 0.1 Hz.

Much more information is provided in the supplementary.
Additionally, the following sentence has been implemented:

The Δf_{rep} fluctuations are happening on the time scale 10-100 ms (see Fig. 1 supplementary).

6 from Reviewer 2

This comment was also addressed in the manuscript.

The huge increase in SNR and cleaning up of the spectrum is due to the relatively narrow window applied during apodization.

Additionally, we have implemented the following abbreviations SNR, $\text{SNR}_{\text{time-correct}}$, $\text{SNR}_{\text{time-corr., apod}}$ to avoid any misunderstandings.

7 from Reviewer 2

This new version of the manuscript includes a measurement on a real sample, Acetylene gas. Moreover, the measurement was apodized and agrees very well with the Yokogawa optical spectrum analyzer. Additionally to it, the following sentences were implemented in the manuscript:

In practical dual-comb applications though, different sample types (gas, solid or liquid) and different spectral resolution can be of interest, which determine the duration of the apodization window. The ideal dual-comb system would provide delays between interferograms on the order of these response times, rendering this noise suppression scheme unnecessary. Ultimately, the spectral resolution is going to define the time window of interest. For example, in liquids (or even solids) the lines are broadened to around 100 GHz which makes it easy to record within approximately 5 μ s duration of the interferogram. Gas at atmospheric pressure with about 1 GHz broadened linewidth can be sufficiently sampled during 0.5 ms which corresponds to 1/10th of the time window between the interferograms. Our Acetylene measurement uses a 20 μ s apodization window (see Fig. 8) and reaches the 5.65 GHz resolution of Yokogawa optical spectrum analyzer.

Point 3

The cited references are in the meantime not up to date - displaying the "most recent" reference from 2019. This list has to be updated. A quick literature check brings novel publications realizing single laser DCS systems like

2020: <https://doi.org/10.1364/OE.403072>,

This work was cited in the supplementary information and added into Table 1.

https://doi.org/10.1364/CLEO_SI.2020.SF3H.2

This work was cited in the main manuscript.

The potential applications in the UV should be emphasized with the following works

2020: <http://dx.doi.org/10.3390/rs12203444>

2021: <https://doi.org/10.1364/OE.424940>

Thank you, all those references were included in the manuscript.

Maybe there is even more recent work to be cited, please check.

We decided to cite the following recent paper:

M. Kowalczyk, Ł. Sterczewski, X. Zhang, V. Petrov, Z. Wang, and J. Sotor, "Dual-Comb Femtosecond Solid-State Laser with Inherent Polarization-Multiplexing," *Laser & Photonics Reviews* 15, 2000441 (2021).

REVIEWER COMMENTS

Reviewer #2 (Remarks to the Author):

From the previous version of the manuscript, the authors made significant changes to clarify or revise some of the earlier claims based on the questions from the reviewers. Also, as they'd made their new setup up and running, additional improved data are provided in this newer version to supplement some of their conclusions. There are some additional issues that hopefully could be addressed before the acceptance for publication:

- 1) Though they further explained the use of a narrow window during apodization when claiming the increase in SNR, it is necessary to spell out the exact window size used here to put the increase in SNR in the more relevant perspective (instead of explaining it in the supplement).
- 2) The verification of the quality of the spectroscopy results shown (like Fig. 4, 7 and 8) are based on the comparison to those from an OSA with standard performance. The dual-comb scheme is expected to offer much better resolution than the OSA, yet the deviation from those results are obvious. Therefore, it is recommend that the claims like 'both spectra agree very well' could be revised. BTW, the model # of the OSA should be double-checked, as it seems to be one unheard of.
- 3) The laser demonstrated seems to show fast and relatively quite large fluctuations in its repetition rate difference as shown in the experimental results. As the demonstrations done in this work are mostly simple samples where this fluctuation could be retrieved relatively easily from the interferogram, it is more challenging to deal with interferogram results with much more complex temporal structures when applied to real-world samples, if compared to some previous dual-comb lasers. This could be one shortcoming that the authors could further explain or acknowledge.

Reviewer #3 (Remarks to the Author):

The authors have re-reviewed the manuscript accordingly to the comments of all reviewers, only very few things are still a bit sloppy. After the authors will have eliminated the following two points, I recommend publication in Nature Communications:

Point 1:

- New sentence on page 8:

The huge increase in SNR and cleaning up of the spectrum is due to the relatively narrow window applied during apodization.

“Relatively narrow window” is too unprecise. Give the value in brackets, even if it is the same value later mentioned in the acetylene experiment on page 11. Otherwise the reader only can assume.

Point 2:

Figure 4 in the supplementary: Please choose different intervals on the time axis, the numbers cannot be read.

Dear Reviewers,

Thank you once again for your feedback on our manuscript. We submit our manuscript with the “tracked” changes such that it is easier for you to read and identify the changes. Additionally, new sentences are highlighted in red.

Reviewer #2 (Remarks to the Author):

From the previous version of the manuscript, the authors made significant changes to clarify or revise some of the earlier claims based on the questions from the reviewers. Also, as they'd made their new setup up and running, additional improved data are provided in this newer version to supplement some of their conclusions. There are some additional issues that hopefully could be addressed before the acceptance for publication:

1) Though they further explained the use of a narrow window during apodization when claiming the increase in SNR, it is necessary to spell out the exact window size used here to put the increase in SNR in the more relevant perspective (instead of explaining it in the supplement).

We implemented the window size in the manuscript:

“The huge increase in SNR and cleaning up of the spectrum is due to the relatively narrow window ($2\ \mu\text{s}$) applied during apodization.”

2) The verification of the quality of the spectroscopy results shown (like Fig. 4, 7 and 8) are based on the comparison to those from an OSA with standard performance. The dual-comb scheme is expected to offer much better resolution than the OSA, yet the deviation from those results are obvious. Therefore, it is recommend that the claims like 'both spectra agree very well' could be revised. BTW, the model # of the OSA should be double-checked, as it seems to be one unheard of.

The quality level of your review is impressive. Thank you for noticing it - the model number of the spectrum analyzer was corrected from AQ630 (wrong) to AQ6370.

Our claim “both spectra agree very well” was changed to “both measurements agree well within a reasonable deviation” - so it is clear that there is no “perfect overlap”. Also, the claim “The Fourier transform of this interferogram is shown in Fig. 8b and it agrees well with the measurement performed by our optical spectrum analyzer” was changed to a more neutral formulation: “The Fourier transform of this interferogram and the corresponding measurement performed by our optical spectrum analyzer are shown in Fig. 8b.”

3) The laser demonstrated seems to show fast and relatively quite large fluctuations in its repetition rate difference as shown in the experimental results. As the demonstrations done in this work are mostly simple samples where this fluctuation could be retrieved relatively easily from the interferogram, it is more challenging to deal with interferogram results with much more complex temporal structures when applied to real-world samples, if compared to some previous dual-comb lasers. This could be one shortcoming that the authors could further explain or acknowledge.

The fluctuations of the repetition rate difference seem to be periodic with a frequency of around 5 Hz (25 interferograms * 7.8 ms \approx 200 ms) as shown in the supplement figure 1:

We were not able to identify the reason for these fluctuations yet. Our lab is, unfortunately, not well isolated from other university facilities. The lab relocation is also planned for the beginning of the next year. The magnitude of the fluctuations is rather small, it is 0.14%, we described it in the supplementary part. “The 11.1 μs peak-to-peak fluctuation corresponds to a relative fluctuation of 1.4×10^{-3} , taking into account the 7.8 ms separation between interferograms.” We have acknowledged the shortcoming mentioned by you in the manuscript discussion part: “Additionally, it is necessary to state with regard to the correction of Δf_{rep} fluctuations, that it will be more challenging to deal with interferograms having much more complex temporal structures originating from complex real-world samples”.

Reviewer #3 (Remarks to the Author):

The authors have re-reviewed the manuscript accordingly to the comments of all reviewers, only very few things are still a bit sloppy. After the authors will have eliminated the following two points, I recommend publication in Nature Communications:

Point 1:

- New sentence on page 8:

The huge increase in SNR and cleaning up of the spectrum is due to the relatively narrow window applied during apodization.

“Relatively narrow window” is too unprecise. Give the value in brackets, even if it is the same value later mentioned in the acetylene experiment on page 11. Otherwise the reader only can assume.

We implemented the window size in the manuscript:

*“The huge increase in SNR and cleaning up of the spectrum is due to the relatively narrow window (**2 μs**) applied during apodization.”*

Point 2:

Figure 4 in the supplementary: Please choose different intervals on the time axis, the numbers cannot be read.

The intervals on the time axis in supplement’s figure 4 were changed accordingly.

REVIEWERS' COMMENTS

Reviewer #2 (Remarks to the Author):

The authors' responses and the corresponding changes to the manuscript are satisfactory to me, and therefore, I'd like to recommend the acceptance of this version of the manuscript.